# Walk-in mental health: Bridging barriers in a pandemic

**Ian Wellspring**[ID]*, **Kirthana Ganesh**[ID], **Kimberly Kreklewetz**

University of British Columbia (Okanagan), Kelowna, Canada

* iwellspr@mail.ubc.ca

## Abstract

'Single Session Therapy' (SST) is a service delivery model that seeks to provide an evidence-based, solution-focused, brief intervention within a single therapy session. The stand-alone session affords the opportunity to provide brief psychological interventions while clients await access to longer-term services. The COVID-19 pandemic has adversely impacted individuals' mental health. However, the majority of research has investigated patient mental health within hospital settings and community organizations that offer long-term services, whereas minimal research has focused on mental health concerns during COVID-19 within an SST model. The primary aim of the study was to measure client experiences of a brief mental health service. The nature of client mental health concerns who access such services at various points during a pandemic was also investigated. The current study utilized client feedback forms and the Computerized Adaptive Testing—Mental Health (CAT-MH) to measure client experiences and mental health concerns. Qualitative analysis of client feedback forms revealed themes of emotional (e.g., safe space) and informational support (e.g., referrals). Clients also reported reduced barriers to accessing services (e.g., no appointment necessary, no cost), as well as limitations (e.g., not enough sessions) of the Walk-in clinic. Profile analysis of the CAT-MH data indicated that clients had higher rates of depression before COVID-19 ($M = 64.2$, $SD = 13.07$) as compared to during the pandemic ($M = 59.78$, $SD = 16.87$). In contrast, higher rates of positive suicidality flags were reported during the pandemic ($n = 54$) as compared to before ($n = 29$). The lower reported rates of depression but higher rate of suicidality during the pandemic was an unanticipated finding that contradicted prior research, to which possible explanations are explored. Taken together, the results demonstrate the positive experiences of clients who access a single session therapy.

**Data Availability Statement:** The data underlying the results presented in the study are available from the primary lead author (Ian Wellspring) at iwellspr@mail.ubc.ca. The data cannot be shared publicly as the data is also collected by the developers for the CAT-MH. Should an individual

## Introduction

Approximately 1 in 5 Canadians will experience a mental health issue within a given year [1]. However, roughly 60% of those individuals will first seek mental health support through their family physician rather than a mental health professional [1]. The most common mental health concerns experienced by Canadians include depression, anxiety, and substance use [2,3].

be interested in accessing the data they are respectfully directed to contact a representative from CAT-MH (Cody Brannon) at cbrannan@adaptivetestingtechnologies.com, in addition to contacting the primary researcher. Further, it should be noted that the co-authors re-conducted the analyses (i.e., ran the code) following the steps outlined in the methods section of the paper and obtained similar results. As such, the findings of the current study should be replicated following the provided data analyses steps. Lastly, the authors of the current paper have special access privileges to the data given that the CAT-MH data is shared under the agreement between UBCO and CAT-MH. Specifically, the agreement states that the CAT-MH should be used for clinical purposes within the WIWC and given that the authors are clinicians within the WIWC they have special access to the data that other clinicians who are not integrated in the WIWC have access to. It should also be reiterated that the authors only have access to CAT-MH data related to the WIWC and do not have access to CAT-MH data from other site locations across North America.

**Funding:** The Walk-in Wellness clinic at the University of British Columbia (Okanagan) is funded by the private donors who wish to remain anonymous. The funders had no role in study design, data collection and analysis, decision to publish, or preparation of the manuscript.

**Competing interests:** The authors have declared that no competing interests exist.

Indeed, a recent systematic review listed depression and anxiety amongst the 10 most common reasons an individual will access a primary care centre [4]. Patients may also present to a family medical centre with interpersonal concerns, symptoms of psychosis, or suicidal ideation, amongst others [5,6]. Mental health concerns may also be exacerbated during times of situational and contextual stress [7].

Research has demonstrated that individuals' mental health tends to deteriorate following a pandemic or natural disaster [8–11]. For instance, Esterwood and Saeed (2020) [10] discuss the acute and long-lasting mental health effects that populations experience following an epidemic and/or natural disaster, such as Ebola, Anthrax threat, tsunamis, and hurricanes (among others). In particular, they illustrated that there tends to be an increase in anxiety, anger, uncertainty, and substance use. Further, research has also demonstrated that rates of depression increase following a societal stressful event [12,13]. Of note, COVID-19 has similarly had a demonstrated negative impact on individuals' mental health [14–16].

Globally, there has been increased rates of substance use [17,18], eating disorders [19], and trauma-related disorders [20] during the pandemic. The adverse impact of the COVID-19 pandemic on individuals' mental health has also been documented across the lifespan, including among children, adolescents, and adults [21]. Individuals from equity-seeking and equity-deserving groups disproportionately experienced negative impacts from the COVID-19 pandemic on their mental health [22–25]. In Canada, a survey conducted by the Angus Reid Institute found that approximately 50% of Canadians found their mental health to have 'severely' worsened since the pandemic [26]. Dozois and the Mental Health Research Canada [27] team found similar results, with participants endorsing extremely high levels of anxiety, which had quadrupled (increasing from 5% to 20%) since COVID-19, while the rates of reported depression more than doubled (increasing from 4% to 10%). It should be noted that not only has there been an increase in self-reported levels of anxiety and depression, but there has also been an increase in the frequency of related diagnoses and symptoms. For instance, there has also been a documented increase in symptoms of anxiety [28], as well as diagnoses of anxiety-related disorders (e.g., Generalized Anxiety Disorder, Illness Anxiety, etc.; [29–31]. There has been a similar increase in the rates of diagnoses of depression [32,33]. Notable symptoms related to depression (which are comorbid with other related diagnoses) are suicidal ideation and attempts to end one's own life, which have also seen an increase during the COVID-19 pandemic [34–36].

Consequently, therapists must address the rise in mental health concerns and corresponding demand for their services, while continuing to provide services to clients on their current caseload (whose mental health may also have deteriorated during the pandemic). To maintain continuity of care when in-person therapy might be untenable (and partially in response to social distancing restrictions), the majority of therapists transitioned from treating clients face-to-face to videoconferencing psychotherapy at least some of the time [37,38]. This further highlights the need for innovative methods of service delivery (both virtual and in-person) to assist clients who may face barriers to attending traditional in-office appointments for extended periods of time. To alleviate some of the pressure on physicians, and the medical system more broadly, clinicians and researchers have developed novel models to deliver mental health services in an accessible and efficient manner, including Single Session Therapy (SST.)

SST refers to a planned single-session intervention whereby the focus is a targeted and intentional offering of mental health support [39]. It is typically provided by a therapist who collaborates with a client to assist that client as much as possible in a single session, with the understanding that further help is available should it be required [40–42]. Indeed, SST involves timely establishment of rapport, assessing risk for imminent harm to self/others, establishing collaborative goals, providing therapeutic support within a single session, and offering

resources for additional services [43]. The flexibility of SST has allowed for services to be used with pediatric, adolescent, and adult populations [43–46], as well as within acute care [47] and outpatient centers [48]. SST has been shown to assist with alleviating mental health concerns of clients who are on waitlists for long-term services [49]. Longitudinal research has demonstrated the maintenance of gains from SST at 18 months, suggesting that individuals who do not return for therapy after one session are not unmotivated, but rather experience sufficient improvement that alleviates the need for further visits [41,45]. Further, Barkham et al. (2006) found that up to 88% of individuals who attend one session of psychotherapy have been noted to achieve reliable and clinically significant improvement [50]. While research exploring the quality of client experiences is sparse, a study by Miller (2008) described that clients cited the immediate accessibility/availability of clinicians, decreased wait times, and warm atmosphere as significant benefits of the SST model, while noting that "longer sessions" (i.e., increased time of session) would be more desirable [51].

SST in a walk-in clinic, whereby each therapeutic encounter is *treated as if it were the only one*, has also been illustrated to be a viable and effective method for reducing barriers and increasing accessibility to mental health services [42]. Program evaluations of such services have reaffirmed their efficacy [46,52]. While systematic reviews have indicated methodological limitations that require further exploration, placing SST on the continuum of mental health care can significantly decrease the burden of mental health concerns [41,43].

## Objectives & hypotheses

The purpose of this study was to better understand not only the broad prevalence of mental health concerns among individuals accessing a walk-in mental health clinic (both before and during COVID-19), but also the ways in which the service was perceived by those that sought treatment. The primary objective was therefore to conduct a qualitative and exploratory analysis of the client feedback data to garner information about client experiences of accessing an SST (i.e., 1–3 sessions) walk-in therapeutic service focussing on whether clients found the service to be effective, as well as the benefits and constraints of the service. Further, it is acknowledged that situational and contextual factors impact how services are accessed, as well as who accesses them. A second objective was therefore to utilize CAT-MH data to measure the nature and severity of mental health concerns of clients who accessed a walk-in mental health clinic before and during the COVID-19 pandemic. Given that the literature has illustrated an increased prevalence of mental health concerns during the pandemic (e.g., 25), it was hypothesized that the general CAT-MH mental health profile would be significantly elevated during the COVID-19 pandemic compared to before the pandemic. In particular, it was hypothesized that the depression, anxiety, suicidality, and substance use subscales would have significantly higher elevations during the COVID-19 pandemic.

## Methods

### Participants

The sample consisted of community members and university students, staff, and faculty who accessed a university mental health clinic in Kelowna, British Columbia: The Walk-In Wellness Clinic (WIWC). The WIWC is a free mental health service within the Psychology Department's Interprofessional Clinic at the University of British Columbia (Okanagan; UBCO). Services are provided by Ph.D. level graduate students in clinical psychology under the supervision of a registered psychologist. The WIWC sessions are brief (30 minutes or less), targeted, and solution-focused. In line with the practice of SST, clients are offered the option of a maximum of 3–4 sessions, but each session is treated as a standalone with no expectation for future

attendance. However, the number of sessions was adjusted if extenuating circumstances were present (e.g., suicidal ideation or other forms of imminent risk). For the present study, only data from the client's first appointment was collected and analyzed. It should be noted that the WIWC was not designed to provide long-term therapy, which is available through other services offered within the Interprofessional Clinic.

Services offered at the WIWC are solution-focused, evidence-based, and primarily utilize a Cognitive-Behavioral-Therapy (CBT) approach. Examples of resources provided to clients include psychoeducation about depression, anxiety, stress, self-care, substance use, and interpersonal issues, along with handouts describing specific techniques to manage these concerns such as grounding, relaxation exercises, goal-setting, communication strategies. Information about long-term mental health services and other relevant community or campus resources is also regularly offered. The WIWC provides short-term solution-focused mental health support to individuals who are often referred by family physicians, walk-in medical clinics, or other community organizations. Its method of service-delivery also aligns with the principles of SST. Clients who access the clinic primarily reside in the Okanagan region of British Columbia; however, clients from anywhere in British Columbia are able to access the service. Since November 2018 the time of the current paper, the clinic has provided mental services to approximately 673 clients. Approximately 60% of clients have been university students, while the remaining 40% have been community members.

## Measures

**Feedback forms.** Clients who access the WIWC are asked to complete a brief feedback form following either their virtual or in-person session. Clients are first asked to describe their primary reason for accessing the WIWC in an open-ended format response option. Second, clients are asked a) how helpful they found the information provided in the session, b) the extent to which they feel their concerns were addressed, and c) how satisfied they were with the session. Response options to these questions are on a Likert scale that only includes two anchors ranging from 0 ("*not at all*") to 100 ("*completely*"). The feedback forms were utilized to assess clients' level of satisfaction with the mental health services, as well as code specific comments made about their levels of satisfaction. Clients are further prompted by two open-ended prompts, which include 'Why would you (or not) recommend the service?' and 'General comments about the service.'

**CAT-MH.** To maximize efficiency, service providers have utilized brief screening questionnaires to provide an overview of a client's presenting concerns and levels of distress [53,54]. The Computerized Adaptive Testing–Mental Health (CAT-MH) is an adaptive measure that queries a client's level of mental health symptoms, including depression, anxiety, trauma, suicidal ideation, mania/hypomania, psychosis, and attentional concerns [55,56]. Notably, the CAT-MH generates different questions based on an individual's answers to previous questions. As such, an individual's initial item responses are used to determine a provisional estimate of their standing on the measured trait (e.g., depression). These responses and their relative severity are used to determine subsequent item selection. The CAT-MH also includes a subscale that utilizes items from the Columbia Suicide Self Rating Scale (C-SSRS) [51] to produce a "suicidality flag," which indicates that suicidality should be further explored by the clinician during the session. The outcome indicates whether suicidality is "present" or "not present." Clients who access the clinic routinely complete the CAT-MH as part of the intake procedure, as well as an anonymous feedback form post-session. The CAT-MH is reviewed by clinicians prior to beginning their session in order to provide information about the potential client concerns and symptom severity.

The CAT-MH has been utilized in outpatient community programs [57], inpatient hospital settings (i.e., emergency rooms) [58], family physician centres that offer mental health services, correctional centres [59], and most notably in primary care settings [60]. The CAT-MH demonstrates high reliability, with test-retest reliability averaging .92 [55,56,61] and high specificity in assessing for potential diagnoses, with Areas Under the Curve (AUC) ranging from .76 for substance use [62] to .94 for depression and PTSD [63]. Measures of specificity for the CAT-MH are similar to other 'gold standard' measures of mental health symptomatology (e.g., Beck Depression Inventory; [55]). Similar findings have been found for the validation and reliability of the CAT-MH to assess for recent substance use [55,62], psychosis [64], suicidal ideation [65], mania/hypomania [57,66], anxiety [56], and post-traumatic stress disorder [63]. It should be noted that the CAT-MH only *indicates the potential presence* of symptoms that may indicate a potential diagnosis, and it is *not* used to provide a diagnosis to individuals.

## Procedure

The current study was a mixed-methods archival data review. Client responses to the anonymous feedback forms were compiled into a Microsoft Excel document. The primary researchers were granted access by the clinic administrator, and the data from the feedback forms were only coded within the clinic (i.e., the raw data was not transported outside of the clinic). As described above, the feedback forms only request general information about the client's perceptions of the therapeutic session (i.e., general level of satisfaction). As such, identifiable and confidential information as to what was specifically discussed during the session was not collected. It should also be noted that clients only completed one feedback form, even if they accessed the service on multiple occasions.

Client responses for those who completed the CAT-MH were also collected, which included the overall subscale scores. The CAT-MH produces an overall severity rating for each subscale that ranges from 0–100. If a client accessed the service multiple times, the CAT-MH scores from the first session was used for analysis, as it was likely the most reflective of their severity of mental health concerns that led them to seek support. each client response is connected to a unique and anonymous (i.e., no identifying information) service provider ID. The CAT-MH data was therefore extracted through collaboration with the CAT-MH developers who provided a Microsoft Excel spreadsheet of the severity ratings associated with the unique account ID for the WIWC. The Excel output provided by the developers was then screened by the primary researchers for the necessary information (e.g., overall severity score).

Neither the feedback forms nor the CAT-MH data collected identifying information (i.e., name, age, gender identity, etc.) are dated. The policies and procedures of the WIWC do not allow for the collection of this information for screening questionnaires and feedback forms in an effort to maintain patient confidentiality, privacy, and anonymity. Therefore, demographic information was not available to be analysed or discussed for the current study. This study was approved by the UBCO Behavioural Research Ethics Board (BREB), approval number H22-00967. Clients provided consent for research during the initial informed consent procedure when accessing services.

## Analysis plan

**Qualitative data.** Using NVivo 12, the responses to the open-ended prompts in the feedback forms were compiled, coded, read, and re-read to identify any themes or patterns that emerged. In line with Braun and Clarke [67] the data were then categorized into themes and sub-themes. The themes were then labelled to reflect content, resulting in four themes and eight sub-themes. A total of 216 forms were analysed. As the feedback forms do not record

session dates they could not be sorted into different phases of the COVID-19 pandemic. Feedback forms were voluntary for clients to complete, and as such they were not always completed following their session, resulting in a smaller size as compared to the CAT-MH data. All available feedback forms were analysed.

**Quantitative data.** Available client CAT-MH data ranged from November 2018 to June 2022. Given the onset of the COVID-19 pandemic, a variable was created to denote whether the client had their session before the COVID-19 pandemic (i.e., before March 11, 2020) or during the pandemic (i.e., after March 11, 2020). After the CAT-MH data was collected, a profile analysis was conducted to determine whether there was a significantly different profile of scores before the COVID-19 pandemic compared to during the pandemic. Follow-up tests included independent samples *t*-tests utilizing the CAT-MH severity scores as the outcome data, while the grouping variable was the aforementioned variable delineating the pandemic timeline (i.e., before or during the COVID-19 pandemic). Wilcoxon Ranked Sum tests or Welch Two-Samples tests were used if the data violated the homogeneity of variance or normality assumptions.

## Results

### Demographics

A total sample of 693 client records were initially reviewed. Approximately 416 clients (60.01%) were UBCO students, staff, or faculty. The remaining 277 clients (39.97%) were community members, primarily from the BC interior region. With regard to the COVID-19 pandemic, approximately 285 clients accessed the clinic prior to March 11, 2020 while 408 accessed the clinic after March 11, 2020. The WIWC began offering virtual services to clients after March 11, 2020 in order to continue services during the pandemic. However, due to the data collection processes, it is unclear how many clients accessed the clinic virtually or in-person after March 11, 2020. As discussed, demographic information (i.e., age, gender identity, cultural identity, etc.) was not available to review for the present study.

### Objective one—General client feedback

Clients were asked to rate their overall level of satisfaction with their session, as well as the extent to which they felt their session was helpful and addressed their concerns. Clients were given the option to complete a feedback form, and at times only completed portions of the form. As such, there were fluctuating sample sizes for each response on the form due to missing data. In general, clients reported that they felt satisfied by the services that they received through the WIWC (*M* = 88.22, *SD* = 14.47; see Table 1).

### Qualitative analysis

Qualitative analysis was conducted utilizing client responses to the open-format response options on the client feedback forms. The analysis revealed four primary themes, which included (1) emotional support, (2) informational support, (3) logistical considerations, and

**Table 1. Client feedback data.**

| Question | *n* | *M* | *Md* | *SD* | Range |
|---|---|---|---|---|---|
| Found the service helpful | 210 | 83.33 | 85 | 16.17 | 20–100 |
| Found that the service addressed their concerns | 207 | 82.47 | 90 | 17.91 | 10–100 |
| Overall satisfaction | 207 | 88.22 | 95 | 14.47 | 10–100 |

**Fig 1. The theme of emotional support (Level I), with the associated sub-themes (Level II) and supporting quotes (Level III).**

(4) limitations of service. Note that some quotes have been corrected for grammar and punctuation.

**Emotional support.**   Clients described the service as providing a safe space to address mental health concerns that were more immediate or pressing in nature (see Fig 1). For instance, a client noted that it was "*a very welcoming and safe environment and I felt like my issues were addressed with care.*" Clients also expressed that they felt a sense of agency, noting that "*it feels good to feel like you're doing something to help yourself.*" They also were mindful of the short-term nature of the service and described it as a "*great booster for [their] mental health.*"

**Informational support.**   Clients were appreciative of the professional competence of the clinicians at the service, as well as the tools, techniques, strategies, and resources provided during and after sessions "*. . .instead of just listening to you rant about your issues without a solution.*" Another individual elaborated on this idea, noting that "*the clinic provided me with a plan on how to bear my problems.*" A number of clients also expressed that the walk-in service was a great way to "*break the ice*" and was "*a low commitment way to test the water*" in the context of psychotherapy.

**Logistical considerations.**   Clients identified several practical features that made the service desirable (see Fig 2). They noted the fact that it was free was an important consideration, as well as its accessibility. They noted that it was a "*low risk way to get support,*" as it can be "*intimidating to see a counsellor.*" The short wait-times, quick turnaround for appointments, and on-campus location were also cited as important factors. Clients also described the service as filling a need, and stated that it "*fills a gap between high priority cases at Interior Health [Provincial Health Authority] and paying for private practice.*"

**Fig 2. The theme of logistical considerations (Level I), with the associated sub-themes (Level II) and supporting quotes (Level III).**

**Limitations of service.** Clients noted that the short session times (~30 minutes) and number of sessions (~3–4 sessions) were at times insufficient to fully address their mental health concerns (see Fig 3). However, most clients also appeared to be aware of the scope of the service, noting "*longer session times would be nice, but I understand why they are shorter*." Ultimately, the need for the service, as well as the acute need for more comprehensive mental health support in general was best described by two clients who said, "*I wished it was a longer appointment but I am grateful for the time I did get*," and "*I wish there could be a more accessible long-term service.*"

### Objective two—Mental health profiles of clients

As noted above, a variable was created to sort CAT-MH responses into two groups, those that sought services before and during COVID-19. A profile analysis was conducted utilizing the profileR package in R [68], which tests for parallelism across profiles. Parallelism provides an indication as to whether a segment of data within a particular profile is significantly different. A test of parallelism revealed a significant difference between profiles, $T2 = 0.187$, $F = 2.243$ (5, 600), $p = .048$. Independent samples $t$-tests were utilized for follow-up tests, with a Bonferroni correction to protect against Type I error rate resulting in significance being accepted at $p = .0083$. Follow-up tests (see Table 2) revealed that there were higher rates of depression before COVID-19 ($M = 64.2$, $SD = 13.07$) as compared to during ($M = 59.78$, $SD = 16.87$), $t = -3.64$, $df = 571.64$, $p = .00029$. Given the onset of the COVID-19 pandemic, and resultant physical distancing measures, it became relevant to understand if there was a significant increase in symptoms of depression and/or suicidality in the immediate aftermath of the WHO's declaration of COVID-19 as a pandemic. Fig 4 depicts scores on the depression subscale, and the highlighted region refers to the scores obtained by clients who accessed the service in the

**Fig 3. The theme of limitations of service (Level I), with the associated sub-themes (Level II) and supporting quotes (Level III).**

March-April 2020. There were no other significant differences found during follow-up analyses for the other mental health domains.

A chi-square test was also conducted to compare the number of 'flags' for suicidality before and during the COVID-19 pandemic, based on the C-SSRS subscale of the CAT-MH. More individuals were flagged as at-risk for suicide during the COVID-19 pandemic ($n = 54$) as compared to before ($n = 29$), $X^2 = 7.77$, $df = 2$, $p = .021$. Fig 5 illustrates the suicidality 'flags' the emerged among clients as a percentage of the total number of clients seen during the

**Table 2. CAT-MH follow-up analysis using independent samples *t*-tests.**

| CAT-MH Subscale | Type of *t*-test | Degrees of Freedom (*df*)[1] | Pre-COVID Levels | | During COVID Levels | | Test Statistic | *p*-value |
|---|---|---|---|---|---|---|---|---|
| | | | *M* | *SD* | *M* | *SD* | | |
| PTSD | Independent Samples | 631.37 | 50.82 | 15.72 | 50.98 | 16.70 | $t = 0.136$ | .89 |
| Mania-hypomania | Wilcoxon Ranked Sum[2] | 631.25 | 32.86 | 19.92 | 33.21 | 21.19 | $W = 61220$ | .92 |
| Suicidality | Wilcoxon Ranked Sum[2] | 591.55 | 46.27 | 17.13 | 45.88 | 16.47 | $W = 58182$ | .62 |
| Substance Use | Wilcoxon Ranked Sum[2] | 625.81 | 49.55 | 12.17 | 49.32 | 12.75 | $W = 60290$ | .89 |
| Anxiety | Welch Two Sample[3] | 651.2 | 50.93 | 18.49 | 51.26 | 20.69 | $t = 0.22$ | .83 |
| Depression | Welch Two Sample[3] | 571.64 | 64.19 | 13.07 | 59.72 | 16.87 | $t = -3.64$ | .00029[4] |

[1]Given that the CAT-MH is adaptive, some questions were not asked for specific subscales impacting the ability for an overall severity score to be calculated, resulting in varying sample sizes for each subscale.

[2]Wilcoxon Ranked Sum Tests were utilized as the data violated the normality assumption.

[3]Welch Two Samples tests were utilized as the data violated the homogeneity of variance assumption (i.e., unequal variances).

[4]Significance accepted at $p = .00833$ due to a Bonferroni correction to protect against Type I error.

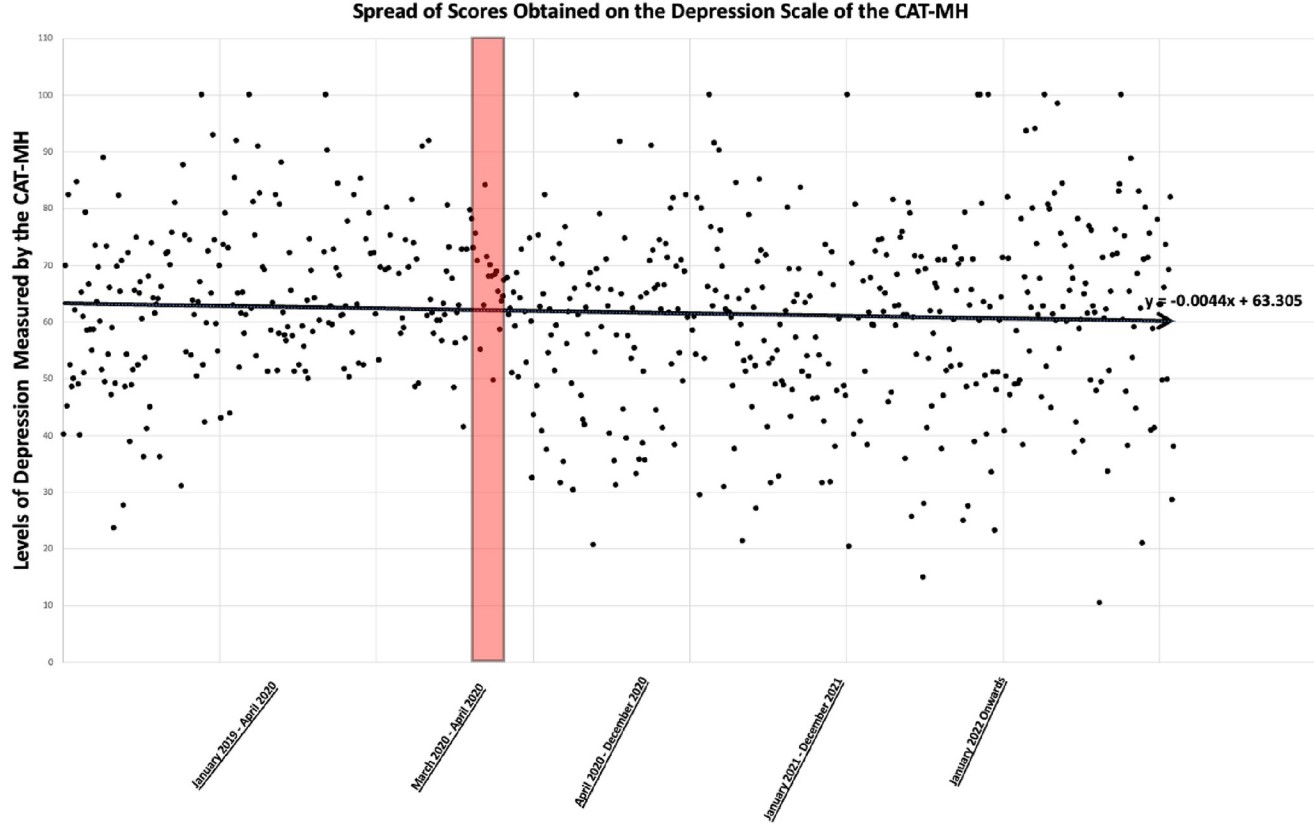

**Fig 4. Scatterplot and line of best fit with March 2020 to April 2020 highlighted in red for depression scores.**

specified time periods. It can be seen that while there was a broad increase in the incidence of suicidality 'flags' after the onset of the pandemic, these rates appear to have peaked in the year 2021.

## Discussion

### Client Experiences of the WIWC

As illustrated in Table 1, feedback from clients was overwhelmingly positive in relation to the context of service and its capacity to address their concerns. The gap between needs and accessibility of mental health services is well documented, both worldwide and within Canada [69,70]. While no single mental health service provider or service is able to address every possible mental health concern(s) that clients may be experiencing, a brief and solution focused model is well-positioned to fill the mental health service gap to accommodate client and societal needs [43,70]. Client feedback from the current study reflects this, with many noting that the walk-in service allowed them to "test the waters" for therapy, describing the service as a more accessible option. Based on our knowledge and familiarity with local mental health resources, it is likely that mental health support would not otherwise be as accessible to clients seen at the WIWC alternate avenues (e.g., the Health Authority or private providers), particularly those who accessed the clinic during the pandemic.

The WIWC is funded largely through private donors, which allows clinicians to provide free services to the community. This feature was directly discussed in clients' decisions to use,

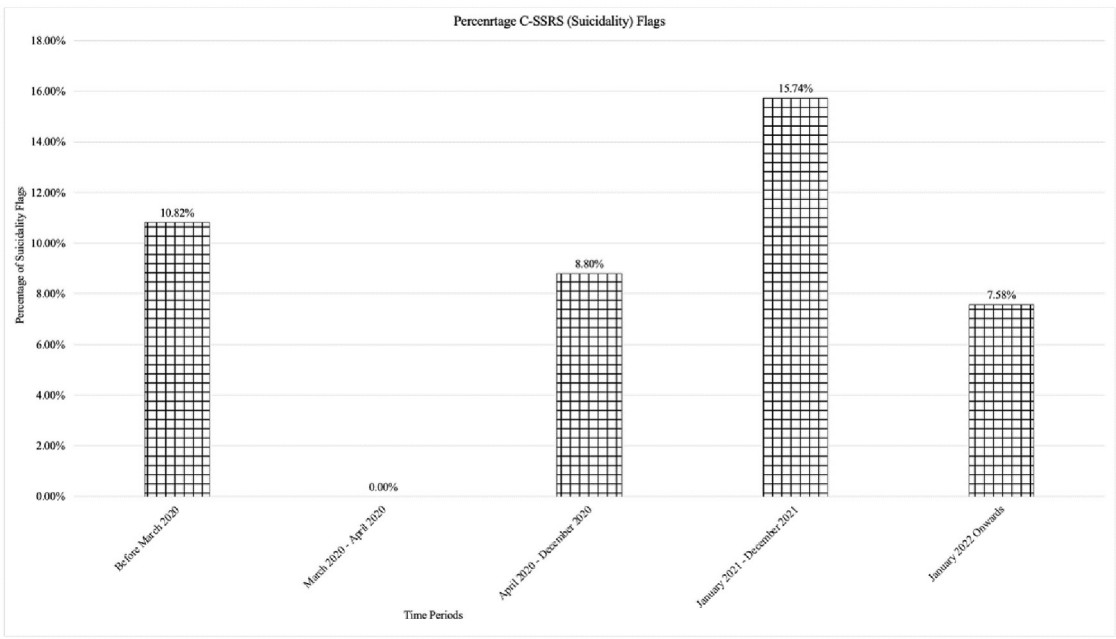

**Fig 5. Bar plot depicting the percentage of suicidality flags per given time period.**

return, and refer the service to others. Research has demonstrated patient reluctance to seek specialist referrals for behavioral health concerns, and a strong preference for such concerns to be addressed within a primary care setting [41,71]. Further, patient comfort with discussing behavioral health concerns, as well as the perceptions that their needs were being met, influenced their opinions of SST services [41,42,71]. The present study corroborated these findings, and clients noted that the structure of a typical session, wherein clinicians offered them a 'menu of options,' allowed clients to choose the focus of the session, themes to discuss, and resources or tools to explore. This influenced a sense of agency, and a sense of comfort in knowing that their concerns were being heard and their needs were being addressed in a way that was meaningful to them. When initial interactions with mental health services are positive, client expectations are enhanced, which has been shown to predict outcome of psychotherapy [72,73]. Further, the modification of the service to adopt a hybrid model due to COVID-19 likely enhanced the capacity to provide services to individuals with chronic health conditions, mobility issues, those from rural areas, and from other marginalizing circumstances, which is an important consideration for mental healthcare in Canada [69]. Ultimately, the qualitative analysis revealed themes that speak to the feasibility of the model, as well as fostering an openness to further mental health services by aiming to reduce mental health stigma and providing psychoeducation about mental health (i.e., how to access services, what therapy 'looks' like, etc.). The potential benefits of integrating SST into the continuum of mental health services include increased access to immediate support, providing patients with meaningful therapeutic interactions at the 'right' moment (when they first seek support), and reduced risks of delayed treatment (or prevent over-treatment, as one session may be sufficient to address certain patients' needs). As such, SST is best-suited for flexible, broad use across diverse clinical settings, which is easy to implement, and requires reduced training to deliver [74], and is likely impacted by those who are accessing the service.

## Mental health before and during COVID-19

It is acknowledged that the experience of an SST model is going to be largely dependent on client individual differences, specifically the mental health concerns that they are experiencing and influencing them to access treatment. Moreover, as discussed, clients' mental health may be impacted or exacerbated by situational and contextual factors that are occurring. As such, in order to provide context for those who were accessing the WIWC clinic, a profile analysis was conducted on data from the CAT-MH, which was split by before and during the COVID-19 given its documented impact on individuals mental health. The profile analysis indicated that there were significantly different mental health profiles based on the CAT-MH questionnaire before the COVID-19 pandemic as compared to during the pandemic. Interestingly, follow-up analysis revealed that participants had higher levels of depression before the COVID-19 pandemic. Indeed, this is in contrast and inconsistent with most prior literature that has illustrated an *increase* in levels of depression during the COVID-19 pandemic [75,76]. For instance, Dozois and the Mental Health Research Canada [27] team found that participants reporting depression had increased from 4% to 10% since COVID-19. However, one prior study found a similar decrease in levels of depression across 11 European countries in older adults [77]. It was suggested that this decrease may be attributable to a sense of perceived 'normalization' of the experience of depression, a global mental health concern due to the COVID-19 pandemic [27,78–80]. Thus, it may have been the case that clients' levels of depression were similar *in relation to* the general population, reducing perceived distress, even if these levels were higher than they had experienced prior to the pandemic. For instance, clients may have compared themselves to their friends or family members who were also experiencing periods of sadness during the pandemic (i.e., thinking 'they're also struggling so I'm not the only one'). Van Winkle and colleagues also noted that clients' lives may have 'slowed down' due to reduced pressure to complete obligations (i.e., work related tasks, social obligations, etc.) in a timely fashion [77]. As such, a decrease in overwhelm may have similarly reduced perceived distress and levels of depression. Regardless, this is indeed a future research question to consider similar results and what may be influencing the depression scores.

It should be noted that questions on the CAT-MH frequently require clients to reflect on the symptoms they have experienced over the past two weeks. Further, requesting clients to reflect on their symptoms in the past two weeks also provides additional context for why they were accessing the WIWC, and their resultant experience. Given that the pandemic has spanned a significant period of time, it is also possible that clients may not have noticed a change in their levels of depression when reflecting on their symptoms over a two-week period. Therefore, if questions were phrased to consider levels of depression over longer periods of time, or if clients were queried to compare their symptoms to before the pandemic, it is more likely that an increase in depression symptomatology would be observed.

The current study also found a higher rate of 'positives' during the COVID-19 pandemic as compared to prior on the C-SSRS indicating that clients were experiencing suicidal ideation, which required further exploration by the clinicians. Indeed, the illustration in Fig 6 further demonstrates the increase in suicidality as a proportion of the clients who accessed clinic services following COVID-19. This finding is in line with prior research that has investigated levels of suicidality that clients have experienced [81,82]. For instance, in a meta-analysis of 54 studies, Dubé and colleagues [81] demonstrated that in comparison to prior to the COVID-19 pandemic, there was a higher incidence of suicidal ideation (10.81%) and attempts to end one's own life (4.68%) during the pandemic. McAuliffe and colleagues [82] also discussed how social and psychological distress as a result of the COVID-19 pandemic likely influenced an increase in suicidal ideation. For instance, the authors noted that financial distress [83],

**Fig 6. The theme of informational support (Level I), with the associated sub-themes (Level II) and supporting quotes (Level III).**

interpersonal difficulties (i.e., decreased social connection; 84), as well as increased substance use and decreased ability to rely on previously developed coping skills (e.g., going to a mall, restaurant, or gym), were factors that may have particularly influenced individuals to experience increased suicidal ideation.

It appears then, that there are apparently paradoxical results—despite lower levels of depression being reported during COVID-19, there was a higher rate of suicidality flags. It is possible that the ease of use and access to mental health services through the WIWC may have enabled clients to feel more comfortable to disclose and discuss suicidal ideation. The WIWC began offering virtual services during the COVID-19 pandemic, which provided the opportunity for clients located anywhere in BC to access services and subsequently increasing access to mental health services. Research has provided initial evidence that clients may feel more comfortable disclosing sensitive information (such as suicidality) over videoconferencing compared to face-to-face communication [85,86]. Glass and Bickler [85] surveyed registered therapists who endorsed finding that their clients were more comfortable disclosing information in videoconferencing psychotherapy. They suggested that this could be related, in part, to clients being in a more comfortable and familiar environment (i.e., at home) which facilitated client disclosure. It is also important to note and reiterate that the suicidality flag is binary, and does not distinguish between ideation, intent, and plan for suicide, and therefore only indicates an area for the clinician for further query. Further, the COVID-19 pandemic may have resulted in periods of reduced responsibility and time engaging in social (i.e., going to a restaurant with friends) and occupational (i.e., going to work) tasks, particularly in the early months. As such, clients who were experiencing thoughts of suicide could have seen an opportunity to access mental health services given the decreased presence of social commitments, and

increased time available to access support. This was supported by the qualitative analysis, which reiterated that ease of access was a significant aspect of client satisfaction.

Another reason for the results may be due to how the CAT-MH is set up to measure symptom severity. The suicidality flag is a binary decision on the C-SSRS, whereas the depression subscale provides a more nuanced severity rating. The primary item for the C-SSRS is whether a client has *ever attempted* to end one's own life. This means that the presence of a historical attempt at any point in their life would automatically register as a suicidality 'flag,' but if the individual does not endorse symptoms of depression *at the time of intake*, their overall severity rating would be lower. Therefore, the likelihood of an individual with a suicidality 'flag' despite lower levels of depression is conceivable.

Taken together, the CAT-MH profile analysis broadly demonstrates that those who accessed the WIWC were continuing to experience anxiety, depression, and symptoms related to a history of stressful life events. Furthermore, there were clients who were also experiencing symptoms of mania and substance use, in addition to more acute symptoms such as suicidal ideation. Given the unique context that the WIWC operated (i.e., during the COVID-19 pandemic) it is interesting to note that general mental health concerns remained consistent, with the exception of depression. Indeed, it could be the case that either alterations or stability in one's mental health concerns likely influence how one experiences an SST-based service. However, this should be considered cautiously and is an area for future research.

## Limitations & future directions

The current study is not without limitations, which themselves suggest directions for future research. First, the present study did not collect demographic information in an effort to maintain client confidentiality and anonymity. As such, there is minimal information available to discuss any demographic factors (i.e., gender identity, age, cultural identity) that may have influenced either the CAT-MH profiles or how individuals experienced the services. This also impacts the ability of the current study to generalize findings to other populations and services. However, given that the WIWC is in a Western and North American context, the present findings can likely be generalized to similar contexts. Future studies may wish to collect demographic information and maintain that data in a separate location to ensure that client confidentiality is still adequately maintained. Relatedly, data regarding mode of service delivery (i.e., virtual or in-person) during the COVID-19 pandemic was not available to be analysed. Therefore, it is also unclear whether the CAT-MH profiles, or client satisfaction, would have varied based on the modality of the session. Indeed, future research may investigate the mental health profiles for clients that tend to access/prefer virtual or in-person services. Perhaps more importantly, future studies may wish to consider asking clients about their preferences, as well as their experiences, of virtual and in-person services.

Second, the current study was limited to client feedback information in open-format responses on a brief client satisfaction survey. Indeed, this resulted in meaningful and integral information related to accessing a mental health service delivered in a non-traditional format (i.e., brief and limited number of sessions, virtual option) SST format. However, it is recognized that perhaps more relevant and detailed information could have been acquired with follow-up focus groups or interviews. In particular, this would have provided an opportunity for interviewers to ask follow-up questions based on the client's initial responses. For instance, it may have elicited more information about the reduced barriers that allowed clients to access the services, as well as any further barriers they experience in accessing care. Future studies investigating SST may seek to collect data initially in a confidential survey format and then

provide an option for respondents to take part in a focus-group or interview (led by individuals other than the clinicians) to provide elaboration on their responses.

Finally, the long-term effects of the COVID-19 pandemic on individuals' mental health remain unknown, particularly given that the COVID-19 pandemic is still ongoing. As such, future studies may seek to conduct long-term evaluations of clients' mental health profiles. Indeed, a longitudinal design would aid in discerning the potential long-term impact of COVID-19 on an individual's mental health. In line with the above, future researchers may also seek to collect long-term data about how mental health services modelled on SST can aid in addressing individuals' mental health concerns, particularly during times of stress (e.g., pandemic). The current study results support the implementation of SST, of which the long-term costs and benefits would need to be further investigated.

## Conclusion

The current study significantly contributes to existing literature by providing unique insight into clients' responses to a brief, time-limited, mental health service operating from a walk-in (or virtual "walk-in") model. Clients reported feeling satisfied with the walk-in service and endorsed feeling emotionally and informationally supported. Clients also reported the logistical considerations that aided in accessing a mental health service, as well as the limitations of the service. Indeed, the present results demonstrate that services based on single-session or limited number of sessions can be effective, from a client perspective, as they consistently reported finding the service useful. The findings also suggest that single-session therapy can be an accessible and appealing alternative service modality for many clients. Indeed, research has highlighted the gap in mental health services that clients experience. As such, it may be feasible for therapists to offer a single session to clients who are on their waitlist in an attempt to provide brief, evidence-based, and solution-focused interventions to manage clients' well-being until such time that they can access long-term services. Alternatively, organizations and health authorities could also more readily consider single-session or brief therapy paradigms to provide services to clients who are on waitlists at other locations, similarly in an effort to assist clients in managing their mental health in the interim.

The current study also investigated clients' severity of mental health concerns before and during the COVID-19 pandemic among clients who accessed a walk-in mental health service. Utilizing the CAT-MH, it was found that the mental health profiles were significantly different for clients accessing a walk-in mental health clinic prior to the pandemic compared to during a pandemic. Interestingly, it was found that there were higher levels of depression prior to the pandemic, which may reflect the 'normalization' of symptoms of depression, or the phrasing of questions on the CAT-MH. It will be important to consider clients' mental health profiles after the COVID-19 pandemic declaration has been discontinued in order to elucidate whether clients' mental health concerns will evolve. Indeed, tracking the long-term impact of a global pandemic on individuals' mental health is a key recommendation not only for researchers, but also for practitioners who provide services, as the changing contextual and situational factors are important to consider when providing therapeutic interventions to clients.

Taken together, the results demonstrate the positive experiences and responses from clients who accessed brief walk-in and / or virtual mental health services. The results further illustrate that a single session/brief therapy service is a viable service delivery model to mitigate the challenges experienced by the healthcare system, and that these models merit further investigation and consideration. The need to also monitor clients' mental health at various time points during major societal events and transitions is also highlighted by the current results. Indeed, the

COVID-19 pandemic is a global phenomenon that will arguably continue to have long-term physical and mental health impacts.

## Author Contributions

**Conceptualization:** Ian Wellspring, Kirthana Ganesh, Kimberly Kreklewetz.

**Data curation:** Ian Wellspring, Kirthana Ganesh, Kimberly Kreklewetz.

**Formal analysis:** Ian Wellspring, Kirthana Ganesh.

**Investigation:** Ian Wellspring, Kirthana Ganesh.

**Methodology:** Ian Wellspring, Kirthana Ganesh.

**Project administration:** Ian Wellspring, Kirthana Ganesh.

**Supervision:** Kimberly Kreklewetz.

**Visualization:** Ian Wellspring, Kirthana Ganesh.

**Writing – original draft:** Ian Wellspring.

**Writing – review & editing:** Ian Wellspring, Kirthana Ganesh, Kimberly Kreklewetz.

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
