## [Decision Letter · Decision Letter 0]

15 Feb 2023

PONE-D-23-00898Mental health during a pandemic: Characteristics and experiences of clients accessing a walk-in mental health clinicPLOS ONE

Dear Dr. Wellspring,

Thank you for submitting your manuscript to PLOS ONE. After careful consideration, we feel that it has merit but does not fully meet PLOS ONE’s publication criteria as it currently stands. Therefore, we invite you to submit a revised version of the manuscript that addresses the points raised during the review process.

We look forward to receiving your revised manuscript.

Kind regards,

Yaara Zisman-Ilani

Academic Editor

PLOS ONE

Journal Requirements:

2. Thank you for stating in your financial disclosure: 

"The Walk-in Wellness clinic at the University of British Columbia (Okanagan) is funded by the private donors who wish to remain anonymous. The funders had no role in study design, data collection and analysis, decision to publish, or preparation of the manuscript"

PLOS ONE requires you to include in your manuscript further information about the funder so that any relevant competing interests can be assessed. Please respond to the following questions:

a. Please state whether any of the research costs or authors' salaries were funded, in whole or in part, by a tobacco company (our policy on tobacco funding is at http://journals.plos.org/plosone/s/disclosure-of-funding-sources)  

b. Please state whether the donor has any competing interests in relation to this work (see http://journals.plos.org/plosone/s/competing-interests) . 

c. Please state whether the identity of the donor might be considered relevant to editors or reviewers’ assessment of the validity of the work.

d. If the donors have no perceived or actual competing interests, please state: “The authors are not aware of any competing interests”. 

This information should be included in your cover letter. We will amend your financial disclosure and competing interests on your behalf.

"The Walk-in Wellness clinic at the University of British Columbia (Okanagan) is funded by the private donors who wish to remain anonymous. "

"The Walk-in Wellness clinic at the University of British Columbia (Okanagan) is funded by the private donors who wish to remain anonymous. The funders had no role in study design, data collection and analysis, decision to publish, or preparation of the manuscript"

Additional Editor Comments:

Dear Dr. Wellspring,

Thank you for your submission to PLOS ONE.

Reviewers have now commented on your paper. The manuscript requires major and significant revisions. If you are prepared to undertake the work required, we would be pleased to reconsider sending your revised manuscript for a review. For your guidance, the reviewers' comments are attached. If you decide to revise and resubmit the manuscript, please include a detailed cover letter explaining which specific changes you made and which recommendations you did not follow and why. This letter should address all of the points raised by each reviewer.

Once the paper has been revised, submit it through the manuscript submission portal.

Sincerely,

Yaara Zisman-Ilani

Reviewers' comments:

Reviewer's Responses to Questions

**Comments to the Author**

1. Is the manuscript technically sound, and do the data support the conclusions?

Reviewer #1: Partly

Reviewer #2: Partly

2. Has the statistical analysis been performed appropriately and rigorously? 

Reviewer #1: Yes

Reviewer #2: Yes

3. Have the authors made all data underlying the findings in their manuscript fully available?

Reviewer #1: No

Reviewer #2: Yes

4. Is the manuscript presented in an intelligible fashion and written in standard English?

Reviewer #1: Yes

Reviewer #2: Yes

5. Review Comments to the Author

Reviewer #1: Thank you for the opportunity to review this manuscript that addresses the topic of COVID-19 effects on mental health, which is still a pressing issue that requires more understanding and addressing. The efficacy of IPC/WIWC is also interesting and can offer a solution to existing gaps in mental health services. The mixed-methods approach provides a broad view of clients' experiences. However, I find that the paper presents two completely different subjects (1. Mental health during and before COVID-19; 2. Clients' satisfaction with WIWC/IPC), and although the authors try to link them in the discussion, they are presented as separate through most of the manuscript and also in the study's procedure (since the feedback forms are not dated, COVID-19 associations with clients' satisfaction were not explored). Therefore, I think the authors should consider removing the parts about clients' feedback regarding WIWC (perhaps presenting them in a different paper), and focusing only on mental health before and during COVID-19. This was the most pressing concern, and it caused confusion throughout the manuscript, from the title to the discussion.

Below I provided more thoughts and feedback, which I hope will be of help.

Title:

1. The title of the paper should be reconsidered. After reading it, I expected that the focus would be mental health during the pandemic. I didn't expect such a large volume about IPC itself (especially in the introduction). I recommend rethinking the title so that it better reflects the content of the paper, or as I suggested above, reorganizing the paper so it includes only one of the subjects.

Abstract:

2. The study aims should be included before describing the CAT-MH.

3. The sentence "Profile analysis indicated a significant difference in mental health profiles" should be made clearer and more accurate to the results it describes. It is the first sentence to describe the main result, so it should be clear and simple.

4. The paper reports a surprising finding: lower depression scores and higher suicidality flags. It should be acknowledged and explained (in the discussion, but also in the abstract) that these two findings together are unexpected.

5. Please remove the word "the" in "the private donors" under financial disclosure.

6. In the last sentence, please use the full term "Integrated Primary Care (IPC)" as it is the first time this abbreviation is mentioned.

Introduction:

7. In general, there is a lot of information here about IPC and not enough about mental health during the pandemic. The title and abstract focused on mental health during COVID-19 and then the introduction only describes it in one paragraph after discussing IPC for two pages. This is confusing.

8. In line 112, I suggest replacing "endorsing" with "reporting".

9. In lines 119-121, I think you should mention that therapists' transition to videoconferencing therapy is usually due to government restrictions during the pandemic.

10. Lines 124-152 (describing the CAT-MH) belong in the METHODS>>MEASURES section.

11. Similar to the previous comment, lines 153-164 (describing the WIWC) belong in the METHODS>>PARTICIPANTS, especially since most of the content already appears in the PARTICIPANTS section.

12. I found it confusing that after such a long description of IPCs, the study took place in WIWC, which lacks the "bridge" element between physical and mental health care. Therefore, I suggest removing or narrowing the part describing the benefits of IPC by highlighting the collaboration between the physician and mental health provider and the reduction of stigma.

13. There is a gap between the OBJECTIVES & HYPOTHESES section which focuses on persons' mental health during COVID-19, and the INTRODUCTION which focuses on the efficacy of IPC. To better reflect the objectives, I suggest again narrowing considerably the IPC part in the introduction and representing more fully the effects of the pandemic on mental health (for example, citing studies not just from Canada, adding literature about mental health during previous pandemics). In general, I recommend clarifying what is already known, what the knowledge gaps are and then using this to justify the study rationale.

14. I don't understand how the two objectives are related. It seems to me that these are two completely different subjects: 1) mental health during COVID-19. 2) Clients' satisfaction with the WIWC. Because the feedback forms are without dates and thus don't allow for a comparison between before and during the pandemic, I think it will be better to remove the second objective and focus this paper only on mental health during the pandemic.

Methods:

15. When describing participants and the WIWC, please elaborate more about the nature of the intervention offered. For example, what kind of professionals work there, how many sessions are offered, what kind of therapeutic approaches are used?

16. Please specify the dates of data collection.

17. Please clarify does each provider read the patient's CAT-MH results before the session? Namely, is the CAT-MH used so that providers will know which issues to focus on during the current session?

18. In line 242, the authors use an example of coding overall severity scores. Why does the researcher need to code severity if "the CAT-MH produces an overall severity rating" (line 234)?

19. Regarding feedback forms, for clients who attended the services more than once, how many forms were analyzed? One or more?

Analysis:

20. The description of the form's open-ended questions (lines 269-274) belongs in the MEASURES section. There is a need for clarification because in the MEASURES>>FEEDBACK FORM section the authors write that the form includes one open-ended question and three multiple-choice questions. Then in the ANALYSIS PLAN section the authors write that the form includes other open-ended questions.

21. The authors report having 693 records and 216 feedback forms of these records being analyzed. How was it decided which feedback forms to analyze?

Results:

22. Again, I think the whole part about treatment satisfaction should be removed (and perhaps reported in a separate paper). However, if you stick to it as is, please make sure that throughout the paper the two subjects appear in the same order. For example, you start the introduction and objectives with the COVID-19 issue, but you start the results with the satisfaction issue. This is confusing.

23. In line 292, satisfaction with what? Treatment? Please specify.

24. How do you explain the gap between suicidality scores presented in the table (where during COVID the score is a little lower than the score before Covid), to the difference in suicidality flags according to chi-square?

25. I suggest presenting the results not according to quantitative/qualitative, but according to the two study objectives.

26. In line 391, I suggest deleting "Fig 2 highlights that" and instead starting the sentence with "Clients were appreciative…".

27. In INFORMATIONAL SUPPORT, it would help if the reader knows what kind of techniques/plans were offered by providers. Otherwise, it sounds vague. This relates to comment no. 15.

Discussion:

28. The sub-titles should be broader. For example, instead of CAT-MH >>> MENTAL HEALTH BEFORE AND DURING COVID-19.

29. The inconsistency between the results of this study and other studies regarding depression is not well explained.

30. In lines 423, 425, please delete the word "for" in "for before the COVID-19…".

31. In line 440, the sentence "as well as how the questions are phrased on the CAT-MH" belongs in the next paragraph and not here.

32. The explanation regarding creativity seems far-fetched since the study didn't ask about clients' creative activities during or before the pandemic.

33. Please rephrase lines 459-462. It is not clear enough. For instance, in line 461, "higher" than what?

34. In line 482, what do you mean by "reduced responsibility"? Lines 482-487 don't seem to be related to the subject described in this section.

35. The possibility that people may feel more comfortable disclosing suicide ideation via videoconference is interesting. However, the authors didn't address the dissonance between the two findings reported here (lower depression scores and higher suicidality flags). This should be addressed and attempted to be explained.

36. In CLIENT FEEDBACK (which also should be replaced with a broader title), the first sentence should be removed.

37. In line 493, the sentence starting with "While no single…" is not clear.

38. In lines 505-506, the authors write "…directly influenced their opinions of IPC". The cited paper reports an association, not a causal relationship. Please rephrase.

39. Under LIMITATIONS, when the authors write about the lack of demographic data, I suggest addressing also the difficulty of generalizing or reaching general conclusions.

40. In lines 534-535, I think the PAI and MMPI are not relevant here, as they are both very long and time-consuming.

41. Can you highlight the added contribution of this study to the research literature?

42. Are there specific recommendations arising from these data?

Conclusions:

43. In line 562, please change "within" to "among".

44. I think this section needs more work. Specifically, the two subjects of the study (mental health during and before covid and satisfaction from WIWC) are intertwined here and should be separated. For example, in lines 574-576 the authors write about the effectiveness of services "both before and during COVID-19". However, since the feedback forms were not dated, referring to before and during COVID here is irrelevant.

References:

45. References no. 77,78 are the same reference.

Reviewer #2: The authors present findings from Computerized Adaptive Testing -- Mental Health (CAT-MH) at intake of a outpatient mental health clinic as well as thematically coded findings from open-ended questions and 3 scaled satisfaction questions completed after their (seemingly first, but this is not obvious from description) session both prior to onset of COVID and during COVID (November 2018 to March 11, 2020 and March 11, 2020, to June 2022, respectively). They describe two objections: 1) "to utilize CAT-MH data to measure the nature and severity of mental health concerns of clients who accessed a walk-in mental health clinic before and during the COVID-19 pandemic" and 2) to explore clients experiences with the walk in clinic through qualitative analysis of the open-ended questions (and given what they report, apparently also quantitative analysis of the 1-100 scales).

Regarding (1), they hypothesized that that depression, anxiety, suicidality, and substance use subscales of the CAT-MH would be worse for those presenting after COVID. In contrast, they present the interesting finding that the only means on subscales of the CAT-MH for the pre-COVID group and the during COVID group were not significantly different, with the exception of the depression subscale, where the pre-COVID group scores reflected significantly greater depression subscales.

Regarding (2), they present findings that the mean score (1-100) for finding the service helpful was 83.33, that for finding the service to address the clients concerns was 82.47, and that for overall satisfaction was 88.22--all reflecting high client satisfaction. They also present thematic findings on open-ended questions that clients found the service to be emotionally supportive; that the interventions to include helpful components (which they label "informational support," but for which many themes do not obviously have to do with providing particular types of information [e.g., that this "was a great way to 'break the ice'"); that clients identified "logistical considerations that appear to reflect reduction in barriers to accessing care' and that concerned limitations of the service (e.g., that it was too brief in length).

The authors conclude that "taken together, the results demonstrate the effectiveness and efficacy of services modeled on IPC in reducing barriers to accessing mental health care, both before and during the COVID-19 pandemic. The results further illustrate that IPC is a viable service delivery model to mitigate the challenges experienced by the healthcare system that merits further investigation and consideration. There are other important and integral research areas to consider, including the long-term effectiveness of IPC [i.e., Integrated Primary Care]."

The finding regarding depression is interesting and the exploratory findings are suggestive. However, the paper suffers from several flaws:

1) The intervention is not IPC. The authors briefly note this on p. 8, lines 188-192. There they write: "It should be noted that the WIWC is modeled on the principles of IPC, but it in and of itself is not an integrated primary care clinic as only mental health professionals provide services. However, the WIWC has fostered various relationships within the Okanagan community and actively collaborates with other service providers to ensure ease of services." The authors do not provide a detailed description of the nature of the walk-in clinic and certainly do not operationalize its model robustly. They do note that services were virtual during the pandemic and they do describe the clinic model as follows: "short-term solution-focused mental health support is offered to individuals who are often referred by family physicians, walk-in medical clinics, or other community organizations, despite an absence of colocated care."

However, this does not appear sufficient to warrant the claim that this is sufficiently like IPC or any adaptation of IPC for their findings to be informative about IPC. They note that sessions are 30 minutes in discussion of the thematic analysis, and they also note in that section that typical services are 3-4 sessions. The rationale for this is unclear. Regardless, with a lack of primary care services, it is unclear that it has any relation to integrated care except perhaps (1) trying to expand access, (2) offering brief interventions, and (3) getting referred by primary care. Importantly, an internet search suggests that the clinic is staffed by graduate psychology students, supervised by psychology faculty. It is unclear what role for psychopharmacology is present, and this would appear to be a major difference with many forms of IPC as the primary care physicians (and on some models psychiatric professionals) can offer psychopharmacological interventions. The parameters and criteria for the interventions are also not adequately detailed to assess their claims about efficacy/effectiveness in the conclusion. It would be helpful to know more about what criteria are used for discharge, completion, or referral out to higher levels of care. It would also be helpful to have a discussion comparing their criteria more fully to fidelity IPC models if the authors insist on this comparison.

2) It is unclear why these two objectives relate to each other in a meaningful way for a single paper. Epidemiological considerations during COVID and the "effectiveness and efficacy" of IPC (on which see below) are quite different topics. Neither objective has a robust additional data with which to explore the various explanations that the authors suggest may explain their findings. This may be understandable given limitations of their non-identifiable dataset, but it raises the sense that two different topics don't have any obvious relation. Moreover, there may be some secondary analyses that might be possible with their data. For instance, is there variation in severity of CAT-MH scores across time during COVID or do rates o presentation/referral/intake vary during COVID. It is not clear that the period between March 11, 2020 and June 2022 should be taken as one homogenous mental health period. These findings would be more informative in order to appropriately interpret their CAT-MH findings.

Given that the topics appear different and if additional exploratory analyses are not possible with whatever other data the authors may have on outcomes, the authors might consider dividing this into two briefer studies submitted as brief reports independently.

3) The exploratory components of the study appear to be preliminary assessments for implementation outcomes. A description of whether the 1-100 point scales had greater than two anchors would be helpful. There are a number of validated implementation scales. It is acceptable to use these 1-100 points scales without validated psychometrics rather than validated implementation scales with standardized psychometrics. However, further description of the anchors of these scales (or any comparison where the scales are used elsewhere) would help.

4) There conclusion that "the results demonstrate the effectiveness and efficacy of services modeled on IPC in reducing barriers to accessing mental health care" appears to be overstated for two reasons. (A) The fact that this is the model is unlike IPC as noted above. (B) The exploratory findings reflect an (early phase) implementation study. There is no clinical outcome with which to assess effectiveness or efficacy as standardly understood. Even if we allow leeway in noting that the authors are speaking of "efficacy ... in reducing barriers" some robust operationalization (and likely comparison) would be needed to "demonstrate ... efficacy." There are a variety of ways in which implementation might be measured and implementation outcomes can be compared across groups, but this doesn't appear to be considered.

6. PLOS authors have the option to publish the peer review history of their article (what does this mean?). If published, this will include your full peer review and any attached files.

Reviewer #1: **Yes: **Rotem Rosenthal Oren

Reviewer #2: No

---

## [Author Response · Author response to Decision Letter 0]

12 May 2023

Dr. Yaara Zisman-Ilani

Academic Editor

PLOS ONE

Submission ID: PONE-D-23-00898

Dear Dr. Zisman-Ilani and Reviewers, 

We appreciate you and the reviewers for your precious time in reviewing our paper and providing valuable comments. These fantastic and insightful comments have led to possible improvements in the current version. 

The authors have carefully considered the comments and tried our best to address every one of them. We hope the manuscript after careful revisions meets your standards. The authors welcome further constructive comments if any. 

Below we provide the point-by-point responses. All modifications in the manuscript have been highlighted in red. 

Sincerely, 

Ian Wellspring (M.A.), Ph.D. Student in Clinical Psychology

Department of Psychology

University of British Columbia (Okanagan) 

Email: iwellspr@mail.ubc.ca

Kirthana Ganesh (M.Sc.), Ph.D. Student in Clinical Psychology

Department of Psychology

University of British Columbia (Okanagan) 

Email: kirthana.ganesh@ubc.ca

Dr. Kimberly Kreklewetz (Ph.D.), Lecturer

Department of Psychology

University of British Columbia (Okanagan) 

Email: kimberly.kreklewetz@ubc.ca

Journal Requirements

Response: Thank-you very much and we have updated the formatting.

2. Thank you for stating in your financial disclosure: "The Walk-in Wellness clinic at the University of British Columbia (Okanagan) is funded by the private donors who wish to remain anonymous. The funders had no role in study design, data collection and analysis, decision to publish, or preparation of the manuscript." PLOS ONE requires you to include in your manuscript further information about the funder so that any relevant competing interests can be assessed. Please respond to the following questions:

a. Please state whether any of the research costs or authors' salaries were funded, in whole or in part, by a tobacco company (our policy on tobacco funding is at http://journals.plos.org/plosone/s/disclosure-of-funding-sources)

b. Please state whether the donor has any competing interests in relation to this work (see http://journals.plos.org/plosone/s/competing-interests) .

c. Please state whether the identity of the donor might be considered relevant to editors or reviewers’ assessment of the validity of the work.

d. If the donors have no perceived or actual competing interests, please state: “The authors are not aware of any competing interests”.

This information should be included in your cover letter. We will amend your financial disclosure and competing interests on your behalf.

Response: Thank-you very much and we have updated the information. Neither the research costs nor authors salaries were funded by a tobacco company. The donor does not have any competing interest, and as such the identity would not be considered relevant to the editors or reviewers. The authors are not aware of any competing interests. 

"The Walk-in Wellness clinic at the University of British Columbia (Okanagan) is funded by the private donors who wish to remain anonymous. " We note that you have provided funding information that is not currently declared in your Funding Statement. However, funding information should not appear in the Acknowledgments section or other areas of your manuscript. We will only publish funding information present in the Funding Statement section of the online submission form. Please remove any funding-related text from the manuscript and let us know how you would like to update your Funding Statement. Currently, your Funding Statement reads as follows: "The Walk-in Wellness clinic at the University of British Columbia (Okanagan) is funded by the private donors who wish to remain anonymous. The funders had no role in study design, data collection and analysis, decision to publish, or preparation of the manuscript. " Please include your amended statements within your cover letter; we will change the online submission form on your behalf.

Response: Thank-you very much. We have removed the funding information from the acknowledgement section. The funding statement can remain as it is stated above given that there are no competing interests that we are aware of, nor the funding provided is from a tobacco company or go directly towards the research salaries. 

Response: Thank-you very much for highlighting this information. However, the current data would not be able to be publicly shared for two primary reasons. Firstly, the CAT-MH data is from a third-party service and as such the current authors do not have legal authority to be able to share the data. The data is owned by the developers of the CAT-MH: Adaptive Testing Technologies. In order to access the data, a researcher would need to contact the primary research (Ian Wellspring) who would then need to coordinate with representatives from CAT-MH, as well as directors of the Walk-in Service to ensure that all data sharing legal requirements are being adhered to by both CAT-MH and the host institution (UBCO). Secondly, client feedback data is sensitive data given that some of the information provided by clients directly mentions their client concerns, what was discussed during their confidential session, and/or includes information that would remove their anonymity. In particular, and as mentioned in the manuscript, patient confidentiality is a primary consideration and as such that data would not be available to be publicly shared. 

Response: Thank-you very much and this has been included in the “Procedure” section. 

Reviewers' comments

Reviewer #1: Thank you for the opportunity to review this manuscript that addresses the topic of COVID-19 effects on mental health, which is still a pressing issue that requires more understanding and addressing. The efficacy of IPC/WIWC is also interesting and can offer a solution to existing gaps in mental health services. The mixed-methods approach provides a broad view of clients' experiences. However, I find that the paper presents two completely different subjects (1. Mental health during and before COVID-19; 2. Clients' satisfaction with WIWC/IPC), and although the authors try to link them in the discussion, they are presented as separate through most of the manuscript and also in the study's procedure (since the feedback forms are not dated, COVID-19 associations with clients' satisfaction were not explored). Therefore, I think the authors should consider removing the parts about clients' feedback regarding WIWC (perhaps presenting them in a different paper), and focusing only on mental health before and during COVID-19. This was the most pressing concern, and it caused confusion throughout the manuscript, from the title to the discussion. Below I provided more thoughts and feedback, which I hope will be of help.

Title:

1. The title of the paper should be reconsidered. After reading it, I expected that the focus would be mental health during the pandemic. I didn't expect such a large volume about IPC itself (especially in the introduction). I recommend rethinking the title so that it better reflects the content of the paper, or as I suggested above, reorganizing the paper so it includes only one of the subjects.

Response: Thank-you for the insight! The title of the manuscript has been adjusted to better reflect the study, and the manuscript itself has been re-organized.

Abstract:

2. The study aims should be included before describing the CAT-MH.

Response: Thank-you and this has been updated.

3. The sentence "Profile analysis indicated a significant difference in mental health profiles" should be made clearer and more accurate to the results it describes. It is the first sentence to describe the main result, so it should be clear and simple.

Response: Thank-you and this has been updated to state “Profile analysis of the CAT-MH data indicated that clients had higher rates of depression before COVID-19 (M = 64.2, SD = 13.07) as compared to during the pandemic (M = 59.78, SD = 16.87).”

4. The paper reports a surprising finding: lower depression scores and higher suicidality flags. It should be acknowledged and explained (in the discussion, but also in the abstract) that these two findings together are unexpected.

Response: Thank-you and this has been updated to state “The lower reported rates of depression but higher rate of suicidality during the pandemic was an unanticipated finding that contradicted prior research, to which possible explanations are explored.” Given word count restrictions, a precise explanation is not provided in the abstract but it is elaborated upon in the discussion.

5. Please remove the word "the" in "the private donors" under financial disclosure.

Response: Thank-you and this has been updated.

6. In the last sentence, please use the full term "Integrated Primary Care (IPC)" as it is the first time this abbreviation is mentioned.

Response: Thank-you and this has been updated and/or removed.

Introduction:

7. In general, there is a lot of information here about IPC and not enough about mental health during the pandemic. The title and abstract focused on mental health during COVID-19 and then the introduction only describes it in one paragraph after discussing IPC for two pages. This is confusing.

Response: Thank-you and this has been updated. Specifically, information related to IPC has been significantly redacted, while information related to COVID-19 (and pandemics/natural disasters more broadly) has been expanded.

8. In line 112, I suggest replacing "endorsing" with "reporting".

Response: Thank-you and this has been updated.

9. In lines 119-121, I think you should mention that therapists' transition to videoconferencing therapy is usually due to government restrictions during the pandemic.

Response: Thank-you and this has been updated.

10. Lines 124-152 (describing the CAT-MH) belong in the METHODS>>MEASURES section.

Response: Thank-you and this information has been moved. 

11. Similar to the previous comment, lines 153-164 (describing the WIWC) belong in the METHODS>>PARTICIPANTS, especially since most of the content already appears in the PARTICIPANTS section.

Response: Thank-you and this information has been moved and updated. 

12. I found it confusing that after such a long description of IPCs, the study took place in WIWC, which lacks the "bridge" element between physical and mental health care. Therefore, I suggest removing or narrowing the part describing the benefits of IPC by highlighting the collaboration between the physician and mental health provider and the reduction of stigma.

Response: Thank-you and as mentioned above, information related to IPC has been decreased.

13. There is a gap between the OBJECTIVES & HYPOTHESES section which focuses on persons' mental health during COVID-19, and the INTRODUCTION which focuses on the efficacy of IPC. To better reflect the objectives, I suggest again narrowing considerably the IPC part in the introduction and representing more fully the effects of the pandemic on mental health (for example, citing studies not just from Canada, adding literature about mental health during previous pandemics). In general, I recommend clarifying what is already known, what the knowledge gaps are and then using this to justify the study rationale.

Response: Thank-you and as mentioned above, information related to IPC has been decreased. Further, information has been included discussing mental health during other pandemics/natural disasters and there has been a re-focus on single session therapy.

14. I don't understand how the two objectives are related. It seems to me that these are two completely different subjects: 1) mental health during COVID-19. 2) Clients' satisfaction with the WIWC. Because the feedback forms are without dates and thus don't allow for a comparison between before and during the pandemic, I think it will be better to remove the second objective and focus this paper only on mental health during the pandemic.

Response: Thank-you and we have attempted to provide a clearer connection between the two study objectives in the manuscript.

Methods:

15. When describing participants and the WIWC, please elaborate more about the nature of the intervention offered. For example, what kind of professionals work there, how many sessions are offered, what kind of therapeutic approaches are used?

Response: Thank-you and we have included more specific information relate the professionals in the service, number and extent of sessions, as well as examples of typically utilized tools and resources.

16. Please specify the dates of data collection.

Response: Thank-you and it is stated in the manuscript that data for before the pandemic spans November 2018 to March 11, 2020, while during the pandemic is from March 11, 2020 to July 2022.

17. Please clarify does each provider read the patient's CAT-MH results before the session? Namely, is the CAT-MH used so that providers will know which issues to focus on during the current session?

Response: Thank-you and we have add clarification to note that the CAT-MH is reviewed by service providers prior to starting their session. 

18. In line 242, the authors use an example of coding overall severity scores. Why does the researcher need to code severity if "the CAT-MH produces an overall severity rating" (line 234)?

19. Regarding feedback forms, for clients who attended the services more than once, how many forms were analyzed? One or more?

Response: Thank-you and it has been clarified that only the first feedback form was reviewed from the first time they accessed services.

Analysis:

20. The description of the form's open-ended questions (lines 269-274) belongs in the MEASURES section. There is 

---

## [Decision Letter · Decision Letter 1]

10 Jul 2023

PONE-D-23-00898R1Walk-in mental health: Bridging barriers in a pandemicPLOS ONE

Dear Dr. Wellspring,

Thank you for submitting your manuscript to PLOS ONE. After careful consideration, we feel that it has merit but does not fully meet PLOS ONE’s publication criteria as it currently stands. Therefore, we invite you to submit a revised version of the manuscript that addresses the points raised during the review process.

We look forward to receiving your revised manuscript.

Kind regards,

Yaara Zisman-Ilani

Academic Editor

PLOS ONE

Reviewers' comments:

Reviewer's Responses to Questions

**Comments to the Author**

1. If the authors have adequately addressed your comments raised in a previous round of review and you feel that this manuscript is now acceptable for publication, you may indicate that here to bypass the “Comments to the Author” section, enter your conflict of interest statement in the “Confidential to Editor” section, and submit your "Accept" recommendation.

Reviewer #1: (No Response)

Reviewer #2: (No Response)

2. Is the manuscript technically sound, and do the data support the conclusions?

Reviewer #1: Yes

Reviewer #2: Yes

3. Has the statistical analysis been performed appropriately and rigorously? 

Reviewer #1: Yes

Reviewer #2: Yes

4. Have the authors made all data underlying the findings in their manuscript fully available?

Reviewer #1: No

Reviewer #2: No

5. Is the manuscript presented in an intelligible fashion and written in standard English?

Reviewer #1: Yes

Reviewer #2: Yes

6. Review Comments to the Author

Reviewer #1: I appreciate the authors' work in response to the comments. In particular, the elaboration of the pandemic's impact on mental health in the introduction and the explanations of the outcomes regarding depression and suicidality are now much more comprehensive. However, I still think the link between the two subjects (service satisfaction and mental health before and during COVID) is not strong and doesn't justify including them in the same paper. Below I provide some more specific comments.

Title and Abstract

1. The revisions made in the title and abstract highly improved these sections.

Introduction

2. In line 68, the word "however" should be removed because this sentence doesn't seem to contradict the previous sentence.

3. Lines 100-103 belong in the METHODS section, not the introduction.

4. I still don't understand why the authors describe the service as IPC. Since the service in the current study doesn't include physicians or any kind of multidisciplinary team, I don't see why it is considered IPC. Therefore, I think the paragraphs about IPC (lines 104-126) should be removed

5. Lines 119-125 seem irrelevant to the paper's topic.

6. I think there is still too much information about SST and IPC and the introduction is still highly unbalanced, with two and a half pages about SST/IPC and only one page about COVID-19. It is also confusing that the authors write about SST, IPC, walk-in clinic, and CC. The part about CC (line 109) can be removed from the introduction because it is described again in the PARTICIPANTS section.

7. The introduction of COVID-19 (line 127) is not well connected to the IPC paragraph that precedes it.

8. Personally, I think it would be better if Covid-19 is described first and SST is described afterward, with lines 149-151 serving as the bridge between these two subjects.

Objectives

9. It is unclear how lines 159-163 explain what follows them (the use of the CAT-MH).

Measures

10. Lines 192-196 belong in the CAT-MH paragraph (lines 218-257).

11. The CAT-MH section is a bit long. I suggest narrowing it down.

12. In line 280, "feedback forms are not dated" can be deleted because it is also written in line 293.

Discussion

13. Line 532, "The current study also found a higher rate of ‘positives’.." is missing "during the pandemic compared to before the pandemic".

14. In line 547, "The present study also illustrated that.." should be rephrased because this is not an additional finding but an explanation of the two findings described earlier.

15. The explanation in lines 559-566 is not clear enough. If you mean that clients had more time during the pandemic to seek treatment for their suicidal thoughts, then why only suicidal ideation was higher and not other symptoms as well?

Reviewer #2: The authors have improved the paper with regard to the concerns that both I and the other reviewer raised. However, I think that issues remain.

Major Concern:

1) Both reviewers were initially concerned that the manuscript was addressing to topics that were not obviously related. The authors revision has imporved this, but the rationale for the second objective given the first objective is still not as clear as it could be.

The authors state that the primary objective is exploratory analysis of qualitative feedback from clients about their experience of SST and their perceptions of its efficacy. To link this to the second objective, they write: "Further, it is acknowledged that situational and contextual factors are going to impact how services are accessed, as well as who accesses them. The role of client motivation may mean that those that sought the service may be those that were ‘ready’ for change. It is also acknowledged that those who were able to access the service were likely individuals who had privileges such as a stable internet connection and/or a working computer/tablet/cellphone. As such, a second objective of the present study was to utilize CAT-MH data to measure the nature and severity of mental health concerns of clients who accessed a walk-in mental health clinic before and during the COVID-166 19 pandemic." (Lines 159-166)

If I understand the authors correctly, this simply means something like "A secondardy objective was to contextualize the qualitative findings in regard to the nature and severity of clients' mental health concerns, as assessed by CAT-MH, because severity is known to correlate with length of time in treatment [or perhaps "time to remission" or perhaps both--here any number of citations could be given]." All the other suggestions appear speculation of potential explanations that are more appropriate in some other section (likely the discussion section). However, it's unclear if I understand them right. If I do not understand them right, the the specific claims need to be justified for the methods section.

Of particular note here, their discussion of "readiness for change" and other socioeconomic determinants does not appear to be measured by the CAT-MH. Thus, if those issues are part of the objective, the study cannot in principle achieve this objective.

2) While the author's revision as above gives more clarity to why two seemingly different topics are brought together into a single paper, I don't see that the authors actually "contextualize" anything in the discussion. The discussion is still offered as largely two independent topics. The secondary objective doesn't obviously contribute to contextualization of the first in the discussion. I think that if I am right about the way the authors are trying to relate the two issues, the discussion needs to reflect that this is a goal.

Significant Concerns to Consider Accounting For:

3) I believe the authors should consider whether they could conduct a secondary analysis for variation during the pre- and during-covid periods in (1) rate of appointments over smaller units of time and (2) severity of symptoms over smaller units of time. I understand they are conducting multiple tests and have already conducted a Bonferroni correction. While it is reasonable to set the parameters for initial analysis purposes, I believe that a secondary analysis would shed more light on the author's finding of the decreased rate of depression with concurrent the increased rate of suicidality that they saw during the Covid period. I take it the author's believe (rightly) that this is one of their most intriguing findings, but they could improve their speculations about it in the discussion through secondary analysis of how suicidality, depresison, and other CAT-MH symptomology varied over time.

Two points are important here:

(a) The authors overlook several possible explanations of the apparent paradoxical finding of decreased depression with increased suicidality that variation of symptoms in time would explain. For instance, high depression scores early in Covid during and shortly after lockdown orders could be masked by increased rate of presentation that they report during Covid on the whole. It is possible that most suicdality presented at time periods that coincide with higher depression scores, but that increased rate of presentation watered down the depression score for the Covid period on the whole. If that were the case, the paradox of increased suicidality rate with decreased depression scores would merely be an artifact of the author's approach. Relatedly, the authors note that they began to offer telehealth services during Covid. In my clinic (and those of my colleagues), the no show rate dropped significantly once telehealth was started. Anecdotally, that easier access seemed to have coincided with more individuals with less symptom burden presenting--even though some individuals got much sicker. (My understanding is that others observed the same phenomneon). Likewise, public attention to mental health during the covid period grew vastly, which might also have resulted in a greater presentation rate of individuals that would not otherwise have presented.

(b) In response to my earlier suggestion they merely stated: "we believe that from a mental health perspective, it is reasonable to consider this as a homogenous group in the present study, due to the salience of mask mandates, physical distancing, etc., that were present throughout this time" in light of the fact that "there has been minimal societal change, COVID-19 is still considered a pandemic, etc." First, this hypothesis is testable by the secondary analysis above. Second, it is not clear that societal change was minimal during this time: there were multiple variations in infection rate, death rate, and hospitalization rate during these periods. There was also significant variation (at least in some locations that I'm more familiar with than the author's location) between stay at home order intensity, amount of standard public interaction, and whether or not children/parents could engage in normal school/work/recrational activities.

Minor comments:

4) Single Session Therapy is mentioned at line 77, but the acronym SST is not introduced to the next mention of SST @ line 78. The acronym should be placed after the first mention.

5) The full name, Computerized Adaptive Testing - Mental Health, is used in the abstract, then CAT-MH is used in several parts of the main paper as an acronym before the full name is respicified at line 221. Please spell out the full name at first use.

7. PLOS authors have the option to publish the peer review history of their article (what does this mean?). If published, this will include your full peer review and any attached files.

Reviewer #1: No

Reviewer #2: No

---

## [Author Response · Author response to Decision Letter 1]

2 Jan 2024

Response to Reviewers

Dr. Yaara Zisman-Ilani

Academic Editor

PLOS ONE

Submission ID: PONE-D-23-00898

Dear Dr. Zisman-Ilani and Reviewers, 

We appreciate you and the reviewers for your precious time in reviewing our paper and providing valuable comments. These fantastic and insightful comments have led to possible improvements in the current version. 

The authors have carefully considered the comments and tried our best to address every one of them. We hope the manuscript after careful revisions meets your standards. The authors welcome further constructive comments if any. 

Below we provide the point-by-point responses. Tracked changes have been utilized to highlight the updates and revisions. 

Sincerely, 

Ian Wellspring (M.A.), Ph.D. Candidate in Clinical Psychology

Department of Psychology

University of British Columbia (Okanagan) 

Email: iwellspr@mail.ubc.ca

Kirthana Ganesh (M.Sc.), Ph.D. Candidate in Clinical Psychology

Department of Psychology

University of British Columbia (Okanagan) 

Email: kirthana.ganesh@ubc.ca

Dr. Kimberly Kreklewetz (Ph.D.), Lecturer

Department of Psychology

University of British Columbia (Okanagan) 

Email: kimberly.kreklewetz@ubc.ca

Reviewers' comments

Reviewer #1: I appreciate the authors' work in response to the comments. In particular, the elaboration of the pandemic's impact on mental health in the introduction and the explanations of the outcomes regarding depression and suicidality are now much more comprehensive. However, I still think the link between the two subjects (service satisfaction and mental health before and during COVID) is not strong and doesn't justify including them in the same paper. Below I provide some more specific comments.

Title and Abstract

1. The revisions made in the title and abstract highly improved these sections.

Thank-you for the support and feedback.

Introduction 

2. In line 68, the word "however" should be removed because this sentence doesn't seem to contradict the previous sentence. 

This has been amended. 

3. Lines 100-103 belong in the METHODS section, not the introduction. 

This has been amended. 

4. I still don't understand why the authors describe the service as IPC. Since the service in the current study doesn't include physicians or any kind of multidisciplinary team, I don't see why it is considered IPC. Therefore, I think the paragraphs about IPC (lines 104-126) should be removed. 

This has been amended and increased clarification has been provided. 

5. Lines 119-125 seem irrelevant to the paper's topic. 

This has been amended. 

6. I think there is still too much information about SST and IPC and the introduction is still highly unbalanced, with two and a half pages about SST/IPC and only one page about COVID-19. It is also confusing that the authors write about SST, IPC, walk-in clinic, and CC. The part about CC (line 109) can be removed from the introduction because it is described again in the PARTICIPANTS section. 

The mentioned comments have been addressed and the introduction has been updated to more accurately reflect a balanced perspective on SST and COVID-19 related mental health concerns. 

7. The introduction of COVID-19 (line 127) is not well connected to the IPC paragraph that precedes it. 

This has been amended. 

8. Personally, I think it would be better if Covid-19 is described first and SST is described afterward, with lines 149-151 serving as the bridge between these two subjects. 

This has been amended. 

Objectives

9. It is unclear how lines 159-163 explain what follows them (the use of the CAT-MH). 

This has been amended. 

Measures 

10. Lines 192-196 belong in the CAT-MH paragraph (lines 218-257). 

This has been amended. 

11. The CAT-MH section is a bit long. I suggest narrowing it down. 

The CAT-MH description has been condensed and re-worded for increased clarification. 

12. In line 280, "feedback forms are not dated" can be deleted because it is also written in line 293. 

This has been amended. 

Discussion

13. Line 532, "The current study also found a higher rate of ‘positives’.." is missing "during the pandemic compared to before the pandemic". 

This has been amended. 

14. In line 547, "The present study also illustrated that.." should be rephrased because this is not an additional finding but an explanation of the two findings described earlier. 

This has been amended. 

15. The explanation in lines 559-566 is not clear enough. If you mean that clients had more time during the pandemic to seek treatment for their suicidal thoughts, then why only suicidal ideation was higher and not other symptoms as well? 

There has been increased clarification provided. 

Reviewer #2: The authors have improved the paper with regard to the concerns that both I and the other reviewer raised. However, I think that issues remain.

Major Concern:

1) Both reviewers were initially concerned that the manuscript was addressing to topics that were not obviously related. The authors revision has imporved this, but the rationale for the second objective given the first objective is still not as clear as it could be.

The authors state that the primary objective is exploratory analysis of qualitative feedback from clients about their experience of SST and their perceptions of its efficacy. To link this to the second objective, they write: "Further, it is acknowledged that situational and contextual factors are going to impact how services are accessed, as well as who accesses them. The role of client motivation may mean that those that sought the service may be those that were ‘ready’ for change. It is also acknowledged that those who were able to access the service were likely individuals who had privileges such as a stable internet connection and/or a working computer/tablet/cellphone. As such, a second objective of the present study was to utilize CAT-MH data to measure the nature and severity of mental health concerns of clients who accessed a walk-in mental health clinic before and during the COVID-166 19 pandemic." (Lines 159-166)

If I understand the authors correctly, this simply means something like "A secondardy objective was to contextualize the qualitative findings in regard to the nature and severity of clients' mental health concerns, as assessed by CAT-MH, because severity is known to correlate with length of time in treatment [or perhaps "time to remission" or perhaps both--here any number of citations could be given]." All the other suggestions appear speculation of potential explanations that are more appropriate in some other section (likely the discussion section). However, it's unclear if I understand them right. If I do not understand them right, the the specific claims need to be justified for the methods section. 

We understand and appreciate the reviewer’s comments on this point. We have revised the study objectives section to better clarify the relationship between the two research questions. The study was primarily designed to understand the efficacy of SST as a treatment modality within a walk-in mental health clinic, and we used a mixed methods approach to comprehensively evaluate this. As COVID-19 emerged as a significant global event during this time with far-reaching mental health implications (among others) it became necessary to separate data into pre-COVID-19 and during the pandemic. We have done so with the quantitative data, but as noted, we were unable to do so with the qualitative data. As such, we have re-written portions of the section to clarify this further. 

Of particular note here, their discussion of "readiness for change" and other socioeconomic determinants does not appear to be measured by the CAT-MH. Thus, if those issues are part of the objective, the study cannot in principle achieve this objective. 

We understand the confusion and have removed this line to clarify. 

2) While the author's revision as above gives more clarity to why two seemingly different topics are brought together into a single paper, I don't see that the authors actually "contextualize" anything in the discussion. The discussion is still offered as largely two independent topics. The secondary objective doesn't obviously contribute to contextualization of the first in the discussion. I think that if I am right about the way the authors are trying to relate the two issues, the discussion needs to reflect that this is a goal.

We have modified the text in the discussion section to clarify the relationship between the two hypotheses. 

Significant Concerns to Consider Accounting For:

3) I believe the authors should consider whether they could conduct a secondary analysis for variation during the pre- and during-covid periods in (1) rate of appointments over smaller units of time and (2) severity of symptoms over smaller units of time. I understand they are conducting multiple tests and have already conducted a Bonferroni correction. While it is reasonable to set the parameters for initial analysis purposes, I believe that a secondary analysis would shed more light on the author's finding of the decreased rate of depression with concurrent the increased rate of suicidality that they saw during the Covid period. I take it the author's believe (rightly) that this is one of their most intriguing findings, but they could improve their speculations about it in the discussion through secondary analysis of how suicidality, depression, and other CAT-MH symptomology varied over time.

We understand and appreciate the reviewer’s comments on this point. We have provided graphical representations of the depression and suicide flag related data to provide increased insight. Further, we have highlighted the two months following the COVID-19 pandemic in each graph. We selected this time frame given that the WIWC services were on hold immediately following (i.e., the following 2 weeks) the COVID-19 declaration, and as such an increased timeframe was required to ensure that an accurate sample size was available. We hope that this provides increased insight into the pattern of mental health symptoms that were apparent during this time. 

Two points are important here:

(a) The authors overlook several possible explanations of the apparent paradoxical finding of decreased depression with increased suicidality that variation of symptoms in time would explain. For instance, high depression scores early in Covid during and shortly after lockdown orders could be masked by increased rate of presentation that they report during Covid on the whole. It is possible that most suicdality presented at time periods that coincide with higher depression scores, but that increased rate of presentation watered down the depression score for the Covid period on the whole. If that were the case, the paradox of increased suicidality rate with decreased depression scores would merely be an artifact of the author's approach. 

We appreciate the reviewer’s depth of thought on this point and have endeavored to clarify the apparent paradox in the discussion section. We also have added a graphical representation of the two most salient findings (i.e., the depression scores and suicidality flags), and have segregated them into various time frames. Our preliminary analyses showed that the changes noted in the overall data between the initial two time periods (pre and during COVID) stayed consistent. The service runs 2 days a week, and so we considered a two-month period (i.e., 16 working days). As such, we believe that the way the data has been currently divided for inferential statistics (i.e., profile analysis) is the most appropriate way to explore the research question, with robust statistical testing. 

Relatedly, the authors note that they began to offer telehealth services during Covid. In my clinic (and those of my colleagues), the no show rate dropped significantly once telehealth was started. Anecdotally, that easier access seemed to have coincided with more individuals with less symptom burden presenting--even though some individuals got much sicker. (My understanding is that others observed the same phenomneon). Likewise, public attention to mental health during the covid period grew vastly, which might also have resulted in a greater presentation rate of individuals that would not otherwise have presented.

We agree that increased public attention to mental health during COVID may have played a role in an increase in individuals seeking treatment. As the clinic is a walk-in service, it is difficult to accurately compare no-shows between in-person and virtual appointments, since individuals often simply presented to the service, or called earlier in the day for a same-day appointment. Moreover, the SST model is built on the assumption that each session is stand-alone (i.e., no obligation for future appointments) and it is not a long-term therapy service in which there is an expectation that clients would attend regular appointments. No-show rates were therefore not evaluated, and are not within the purview of the modality of SST. Intuitively, it is reasonable to expect that individuals with less symptom burden may have presented during COVID due to ease of access; however, our results demonstrate that there were no significant differences in the rates of anxiety, substance use, and trauma symptoms, and the only differences were in depression and suicidality, which have explored in the discussion section. 

(b) In response to my earlier suggestion they merely stated: "we believe that from a mental health perspective, it is reasonable to consider this as a homogenous group in the present study, due to the salience of mask mandates, physical distancing, etc., that were present throughout this time" in light of the fact that "there has been minimal societal change, COVID-19 is still considered a pandemic, etc." First, this hypothesis is testable by the secondary analysis above. Second, it is not clear that societal change was minimal during this time: there were multiple variations in infection rate, death rate, and hospitalization rate during these periods. There was also significant variation (at least in some locations that I'm more familiar with than the author's location) between stay at home order intensity, amount of standard public interaction, and whether or not children/parents could engage in normal school/work/recrational activities.

We agree and appreciate the reviewer’s comments to this section. We have altered the wording to this section to provide increased clarification and insight. 

Minor comments:

4) Single Session Therapy is mentioned at line 77, but the acronym SST is not introduced to the next mention of SST @ line 78. The acronym should be placed after the first mention. 

This has been amended.

5) The full name, Computerized Adaptive Testing - Mental Health, is used in the abstract, then CAT-MH is used in several parts of the main paper as an acronym before the full name is respicified at line 221. Please spell out the full name at first use. 

This has been amended.

---

## [Decision Letter · Decision Letter 2]

2 Apr 2024

PONE-D-23-00898R2Walk-in mental health: Bridging barriers in a pandemicPLOS ONE

Dear Dr. Wellspring,

Thank you for submitting your manuscript to PLOS ONE. After careful consideration, we feel that it has merit but does not fully meet PLOS ONE’s publication criteria as it currently stands. Therefore, we invite you to submit a revised version of the manuscript that addresses the points raised during the review process.

I agree with the reviewers' judgments recommending minor revisions. Please attend to the minor revisions noted by the reviewers, and also, please include the period of time from which client data were used in the method section--this was included later in the manuscript but should also be included in the method section, as it highly relevant to contextualizing the data.

We look forward to receiving your revised manuscript.

Kind regards,

Emily Lund

Academic Editor

PLOS ONE

Journal Requirements:

Additional Editor Comments:

I agree with the reviewers' judgments recommending minor revisions. Please attend to the minor revisions noted by the reviewers, and also, please include the period of time from which client data were used in the method section--this was included later in the manuscript but should also be included in the method section, as it highly relevant to contextualizing the data.

Reviewers' comments:

Reviewer's Responses to Questions

**Comments to the Author**

1. If the authors have adequately addressed your comments raised in a previous round of review and you feel that this manuscript is now acceptable for publication, you may indicate that here to bypass the “Comments to the Author” section, enter your conflict of interest statement in the “Confidential to Editor” section, and submit your "Accept" recommendation.

Reviewer #1: (No Response)

Reviewer #2: All comments have been addressed

2. Is the manuscript technically sound, and do the data support the conclusions?

Reviewer #1: Yes

Reviewer #2: Yes

3. Has the statistical analysis been performed appropriately and rigorously? 

Reviewer #1: Yes

Reviewer #2: Yes

4. Have the authors made all data underlying the findings in their manuscript fully available?

Reviewer #1: No

Reviewer #2: No

5. Is the manuscript presented in an intelligible fashion and written in standard English?

Reviewer #1: Yes

Reviewer #2: Yes

6. Review Comments to the Author

Reviewer #1: I appreciate the authors work on the manuscript, which is now more integrative.

Below are some minor issues I suggest to address.

Introduction

• The revisions made in this section are excellent.

Methods

• Line 171 “However, the number of sessions was 172 altered if extenuating circumstances were present (e.g., suicidal ideation or other forms of 173 imminent risk).” - Are there participants in the current study that attended more than 4 sessions due to such circumstances? If so it should be considered whether they should have been removed from analysis because it becomes a different kind of treatment and not necessarily SST.

Discussion

• Lines 469-473: Does this sentence reflect in the results as well? Did participants address this advantage in the open ended questions?

• Lines 536-538: I suggest to remove this because it’s not directly related to the current study.

• Line 606: IPC was removed from the paper so it should be removed from here also.

Reviewer #2: The authors have improved the manuscript significantly. I have only minor, easily addressable comments below.

Substantive comments:

1) P. 33, line 559 says that a hypothesis about increased suicidality resulting from improved ease of access was "substantiated" by the qualitative findings which reflect ease of access. However, the anonymity of the authors' data precludes showing that the individuals with suicidality were the ones who reported increased ease of access. Thus, the verb should be weakened to one that doesn't suggest direct hypothesis testing occurred. "Supported" would be one alternative.

2) P. 35, line 606: The paper is no longer discussing IPC and hence reference to IPC needs to be replaced here.

Minor grammatical comments:

1) P. 13, lines 83-84: "the Covid-19" is awkward. Perhaps either the "the" should be deleted or "pandemic" (or some similar word) should be added.

2) P. 14, line 99: consider hyphenating "anxiety-related disorders."

3) P. 14, lines 101-103: "A notable symptom related to depression (and is comorbid with other related diagnoses) is the..." is not parallel. Considering the phrase after the "and" renders the predicate to read "symptom ... is ... is." Perhaps the "and" is supposed to be a "that."

Additionally, this sentence mistakenly reads that "a ... symptom ... is the rates," which is non-sensical/a category mistake. The authors appear to be meaning to talk about the rates of symptoms rather than saying that the symptom(s) is/are themselves rates.

4) P. 14, lines 114-115: "including, and Single Session Therapy" is ungrammatical. Either the "and" should be deleted or another noun is needed before it.

5) P. 15, line 129: A space is needed after "visit."

6) P. 17, lines 183-184: The phrase "despite an absence of co-located care" is now unnecessary because the prior framing of the paper in terms of integrative care has been changed. In that setting, it is also confusing. Consider deleting.

7) P. 32, line 539: The first sentence of the paragraph is ungrammatical with subject predicate disagreement between "is" and "results."

8) P. 32, line 541: There is a comma splice after "WWIC."

9) P. 35, line 626 et passim: The beginning of the sentence is wordy and awkward: "A specific recommendation that may arise from these findings is for programs and service providers to ..." Consider reducing wordiness.

7. PLOS authors have the option to publish the peer review history of their article (what does this mean?). If published, this will include your full peer review and any attached files.

Reviewer #1: No

Reviewer #2: No

---

## [Author Response · Author response to Decision Letter 2]

7 Apr 2024

Dr. Emily Lund

Academic Editor

PLOS ONE

Submission ID: PONE-D-23-00898

Dear Dr. Lund and Reviewers, 

We appreciate you and the reviewers for your precious time in reviewing our paper and providing valuable comments. These fantastic and insightful comments have led to possible improvements in the current version. 

The authors have carefully considered the comments and tried our best to address every one of them. We hope the manuscript after careful revisions meets your standards. The authors welcome further constructive comments if any. 

Below we provide the point-by-point responses. Tracked changes have been utilized to highlight the updates and revisions. 

Sincerely, 

Ian Wellspring (M.A.), Ph.D. Candidate in Clinical Psychology

Department of Psychology

University of British Columbia (Okanagan) 

Email: iwellspr@mail.ubc.ca

Kirthana Ganesh (M.Sc.), Ph.D. Candidate in Clinical Psychology

Department of Psychology

University of British Columbia (Okanagan) 

Email: kirthana.ganesh@ubc.ca

Dr. Kimberly Kreklewetz (Ph.D.), Lecturer

Department of Psychology

University of British Columbia (Okanagan) 

Email: kimberly.kreklewetz@ubc.ca

 

Reviewers’ Comments

Reviewer #1: I appreciate the authors work on the manuscript, which is now more integrative.

Thank-you for the support and feedback.

Introduction

1. The revisions made in this section are excellent.

Thank-you for the support and feedback.

Methods

2. Line 171 “However, the number of sessions was 172 altered if extenuating circumstances were present (e.g., suicidal ideation or other forms of 173 imminent risk).” - Are there participants in the current study that attended more than 4 sessions due to such circumstances? If so it should be considered whether they should have been removed from analysis because it becomes a different kind of treatment and not necessarily SST.

We appreciate the clarification. All participants completed only one feedback form (on their first appointment). Similarly, only the CAT-MH scores from their first appointment used for the present study, as they were the most representative of the reason why clients accessed the service. This is described in lines 245-246 and 249-251 in the Procedures section of the revised document with track changes enabled. Further, an additional sentence has been added after line 171: “For the present study, only data from the client’s first appointment was collected and analyzed.”

Discussion

3. Lines 469-473: Does this sentence reflect in the results as well? Did participants address this advantage in the open ended questions?

Ease of access was a theme that emerged; however, specific references to virtual service delivery were not present. The word “likely” has been added to this sentence to communicate the extrapolation made by the researchers.

4. Lines 536-538: I suggest to remove this because it’s not directly related to the current study.

The sentence has been removed. 

5. Line 606: IPC was removed from the paper so it should be removed from here also.

The term IPC has been replaced with SST. 

Reviewer #2: The authors have improved the manuscript significantly. I have only minor, easily addressable comments below.

Thank-you for the support and feedback.

Substantive comments:

1. P. 33, line 559 says that a hypothesis about increased suicidality resulting from improved ease of access was "substantiated" by the qualitative findings which reflect ease of access. However, the anonymity of the authors' data precludes showing that the individuals with suicidality were the ones who reported increased ease of access. Thus, the verb should be weakened to one that doesn't suggest direct hypothesis testing occurred. "Supported" would be one alternative.

We appreciate the clarification. The line has been amended.

2. P. 35, line 606: The paper is no longer discussing IPC and hence reference to IPC needs to be replaced here.

The term IPC has been replaced with SST. 

Minor grammatical comments:

1. P. 13, lines 83-84: "the Covid-19" is awkward. Perhaps either the "the" should be deleted or "pandemic" (or some similar word) should be added.

The word “the” has been deleted.

2. P. 14, line 99: consider hyphenating "anxiety-related disorders."

The term has been amended. 

3. P. 14, lines 101-103: "A notable symptom related to depression (and is comorbid with other related diagnoses) is the..." is not parallel. Considering the phrase after the "and" renders the predicate to read "symptom ... is ... is." Perhaps the "and" is supposed to be a "that." Additionally, this sentence mistakenly reads that "a ... symptom ... is the rates," which is non-sensical/a category mistake. The authors appear to be meaning to talk about the rates of symptoms rather than saying that the symptom(s) is/are themselves rates.

Thank-you for the insight. The sentence has been rewritten as follows: “Notable symptoms related to depression (which are comorbid with other related diagnoses) are suicidal ideation and attempts to end one’s own life, which have also seen an increase during the COVID-19 pandemic.”

4. P. 14, lines 114-115: "including, and Single Session Therapy" is ungrammatical. Either the "and" should be deleted or another noun is needed before it.

The word “and” has been deleted. 

5. P. 15, line 129: A space is needed after "visit." 

A space has been added. 

6. P. 17, lines 183-184: The phrase "despite an absence of co-located care" is now unnecessary because the prior framing of the paper in terms of integrative care has been changed. In that setting, it is also confusing. Consider deleting.

The phrase has been deleted.

7. P. 32, line 539: The first sentence of the paragraph is ungrammatical with subject predicate disagreement between "is" and "results."

The word “is” has been changed to “are.”

8. P. 32, line 541: There is a comma splice after "WWIC."

The comma has been removed.

9. P. 35, line 626 et passim: The beginning of the sentence is wordy and awkward: "A specific recommendation that may arise from these findings is for programs and service providers to ..." Consider reducing wordiness.

Thank-you for the insight. The sentence has been reworded as follows: “The findings also suggest that single-session therapy can be an accessible and appealing alternative service modality for many clients.”

---

## [Editor Report · Decision Letter 3]

9 Apr 2024

Walk-in mental health: Bridging barriers in a pandemic

PONE-D-23-00898R3

Dear Dr. Wellspring,

We’re pleased to inform you that your manuscript has been judged scientifically suitable for publication and will be formally accepted for publication once it meets all outstanding technical requirements.

Kind regards,

Emily Lund

Academic Editor

PLOS ONE